# Digital Biohacking Approach to Dietary Interventions: A Comprehensive Strategy for Healthy and Sustainable Weight Loss

**DOI:** 10.3390/nu16132021

**Published:** 2024-06-26

**Authors:** Alessio Abeltino, Giada Bianchetti, Cassandra Serantoni, Alessia Riente, Marco De Spirito, Giuseppe Maulucci

**Affiliations:** 1Dipartimento di Neuroscienze, Sezione di Biofisica, Università Cattolica del Sacro Cuore, 00168 Rome, Italy; alessio.abeltino@unicatt.it (A.A.); giada.bianchetti@unicatt.it (G.B.); cassandra.serantoni@unicatt.it (C.S.); alessia.riente@unicatt.it (A.R.); marco.despirito@unicatt.it (M.D.S.); 2Fondazione Policlinico Universitario Agostino Gemelli, IRCCS, 00168 Rome, Italy

**Keywords:** obesity, digital biohacking, deep learning, sustainable weight loss, carbon footprint, personalized approach, metabolic avatar

## Abstract

The rising obesity epidemic requires effective and sustainable weight loss intervention strategies that take into account both of individual preferences and environmental impact. This study aims to develop and evaluate the effectiveness of an innovative digital biohacking approach for dietary modifications in promoting sustainable weight loss and reducing carbon footprint impact. A pilot study was conducted involving four participants who monitored their weight, diet, and activities over the course of a year. Data on food consumption, carbon footprint impact, calorie intake, macronutrient composition, weight, and energy expenditure were collected. A digital replica of the metabolism based on nutritional information, the Personalized Metabolic Avatar (PMA), was used to simulate weight changes, plan, and execute the digital biohacking approach to dietary interventions. The dietary modifications suggested by the digital biohacking approach resulted in an average daily calorie reduction of 236.78 kcal (14.24%) and a 15.12% reduction in carbon footprint impact (−736.48 gCO_2_eq) per participant. Digital biohacking simulations using PMA showed significant differences in weight change compared to actual recorded data, indicating effective weight reduction with the digital biohacking diet. Additionally, linear regression analysis on real data revealed a significant correlation between adherence to the suggested diet and weight loss. In conclusion, the digital biohacking recommendations provide a personalized and sustainable approach to weight loss, simultaneously reducing calorie intake and minimizing the carbon footprint impact. This approach shows promise in combating obesity while considering both individual preferences and environmental sustainability.

## 1. Introduction

Obesity has emerged as a rapidly growing epidemic, with environmental influences and lifestyle modifications playing a significant role [1,2]. Nutrition and exercise are modifiable factors that exert a substantial impact on energy balance (EB) [3] and weight management. However, despite extensive research, there continues to be an ongoing debate surrounding the energy content and optimal macronutrient distribution for promoting effective and healthy weight loss [4,5]. Low-fat diets [6] have long been recommended to reduce obesity, but their effectiveness has recently come under scrutiny due to the persistent rise in obesity rates despite reductions in fat intake [2,4,5]. Conversely, very low carbohydrate (ketogenic) [7] diets have gained popularity, and several recent clinical trials suggest their superiority in promoting short-term weight loss and improving metabolic syndrome characteristics compared to low-fat diets [2,4,5]. However, these diets cause problems in the long term despite their efficacy in the short term [8]. Previous studies have highlighted potential issues associated with the ketogenic diet in the long term, such as promoting metabolic acidosis and renal hyperfiltration, which may ultimately contribute to a significant reduction in life expectancy, especially in middle-aged individuals [9].

Indeed, making drastic changes to dietary habits is a difficult task that remains challenging [10]. Numerous factors, including personal taste, social life, and work demands, come into play, increasing the difficulty of adhering to one diet over another [11]. Therefore, to achieve lasting results, a comprehensive and lifelong approach is needed, incorporating permanent lifestyle changes that seamlessly integrate into individuals’ lives.

Furthermore, weight loss should not be the sole focus of interventions; instead, considerations should extend to another important parameter that is not commonly considered: sustainability. Indeed, food consumption is among the main drivers of environmental impacts [12] (Appendix A). Therefore, the generation of diets aimed at reducing this impact can be crucial to addressing this problem [13]. 

To overcome these issues, we developed a new dietary intervention based on a digital biohacking approach [14,15], which relies on the use of small, incremental changes to the usual individual diet, allowing a soft transition to a healthier and more sustainable diet. The basic idea is to select foods with the highest calorie intake in each meal and replace them with healthier and more sustainable foods taken from personal food diaries [16,17,18]. In this way, micro changes are made in the person’s lifestyle, minimizing both social and environmental impact by considering parameters such as the carbon footprint [19]. 

For optimal and personalized planning, and to understand individual responses to diet and exercise interventions, we simulated the effect of these dietary interventions by exploiting the Personalized Metabolic Avatar [17,18] (PMA), a digital model based on a Gated Recurrent Unit (GRU) cell that replicates the metabolism by predicting weight changes using nutritional input.

In this pilot study, we aimed to test the feasibility and initial effectiveness of a novel digital biohacking approach to dietary interventions. While acknowledging the limited sample size and demographic constraints, we deliberately selected a small, controlled cohort to rigorously test and refine our methodology. This initial cohort comprised predominantly normal BMI individuals to ensure a controlled baseline for evaluating the algorithm’s precision and adaptability in a sedentary population. Furthermore, focusing on a limited number of participants allowed us to closely monitor and accurately measure a wide array of variables necessary for developing our PMA.

## 2. Materials and Methods

### 2.1. Study Population 

In this single-arm, uncontrolled pilot prospective study (Figure 1), our sample comprised four voluntary, healthy Italian adults with sedentary lifestyles (25% females and 75% males). The mean BMI of participants was 23.42 ± 1.68, and their average age was 44.75 ± 10.23 years. One participant was classified as overweight, while the remaining three had a normal weight status. They were recruited among our lab staff and self-monitored their weight, diet, and activities completed for more than 1 year, daily, as already shown in previous works, under the surveillance of a nutritionist [17,18]. The participants shared their personal data after signing informed consent. 

This specific demographic was chosen to provide a controlled baseline, ensuring that the effects of the digital biohacking interventions could be measured with minimal confounding variables. Although this cohort is not representative of the broader population, it provides valuable preliminary data for refining our methodology. Future studies will aim to include larger, more diverse populations to validate and generalize our findings.

### 2.2. Data Collection

In the generation of biohacked diets, the following data were used:-Food consumption data: Our team developed a web-based application (ArMOnIA, https://www.apparmonia.com, accessed on 25 January 2024) which allowed us to retrieve a comprehensive list of all the food consumed over a period of more than 1 year for each participant. This application facilitated the comprehensive tracking of all foods consumed by each participant over a period exceeding one year. Participants could input their dietary intake during the monitoring period directly into the application, with the data stored in a NoSQL database. This list includes detailed information about the macronutrient composition, calorie intake, and food category of each food item. Furthermore, the foods have been categorized into six meals for each day: the main ones (Breakfast, Lunch, and Dinner) and snacks between them.-Carbon footprint assessment: We conducted a thorough analysis of the carbon footprint impact associated with each food item. To calculate this impact, we utilized a classification system that corresponds to the My Emission-free Food Carbon Footprint Calculator database (https://myemissions.green/food-carbon-footprint-calculator/, accessed on 25 January 2024). Each food was assigned to the respective food class, enabling us to accurately determine its carbon footprint impact.

In addition, to simulate the metabolic response to digital biohacking, we considered various other data sources, which have been extensively described in our presentation of the PMA [17]:Daily calorie intake and macronutrient composition were obtained from ArMOnIA, where users input their dietary information into a structured NoSQL database.Daily weight and Resting Metabolic Rate (RMR) were obtained from the Mi Body Composition Scale 2 [20]. Users weighed themselves each morning before breakfast, with this data accessed via an API integrated into ArMOnIA through an Amazfit Developer Account.Daily energy expenditure for Physical Activities (PA) was collected from MiBand 6 [21]. MiBand 6 was worn 24/7 for the duration of the study, allowing it to be recharged for one hour approximately once a week. These activities were accurately recorded and retrieved through dedicated APIs, as outlined in the previous point.

These data were utilized to calculate the daily EB and the daily macronutrient composition, which serve as inputs for the PMA. To assess the efficacy of digital biohacking, we performed statistical analyses using this data and simulated the effects of different diets using the PMA. Subsequently, we analyzed real-world data to authenticate the effectiveness of biohacking under real conditions.

Missing data are addressed, as already explained in the previous study [17].

### 2.3. Digital Biohacking

Unlike standardized diet plans and low-fat diets, which often lack personalization and show limited effectiveness, the biohacking approach not only focuses on personalized and sustainable dietary modifications but also considers individual preferences and health needs, tailoring dietary changes to individual metabolic responses and potentially leading to more sustainable weight management. Ketogenic diets, while effective in the short term, pose long-term health risks [22], whereas our approach implements small, sustainable modifications that are easier to maintain. Intermittent fasting can be challenging to adhere to for some individuals, but our incremental adjustments offer a more flexible and personalized solution. Unlike other diets that require drastic changes, our biohacking approach builds on the user’s existing diet, making sensible substitutions that lead to gradual, permanent changes in habits and stable weight loss without the yo-yo effect. 

The generation of dietary interventions based on a digital biohacking approach was made by an algorithm considering all food data retrieved in more than 1 year of data collection, as explained in Section 2.2. The digital biohacking algorithm leverages a PMA to simulate individual metabolic responses based on personalized data inputs. The key steps in the algorithm include:Initialization: The algorithm begins by copying a dataset related to dietary habits (diet_week) into another variable (bh). Additionally, empty lists are initialized to store information about replacements (indexes, switches, meals, new, calories_reduction, impact_reduction, quantity), and an empty dictionary (dictionary) is created to track correspondences between replaced and new food items.Data iteration: For each unique date-meal combination, the algorithm evaluates the total calorie intake. If it is below a threshold (e.g., 100 kcal), no replacement is made. Otherwise, the food item with the highest caloric intake (excluding condiments and spices) is identified for replacement. The algorithm searches for alternative foods within the same meal category, aiming to reduce the caloric intake by 100–200 kcal while considering the carbon footprint impact.Food replacement: Suitable alternative food options are selected randomly from a pre-defined list, ensuring they belong to the same macro-category and have similar nutritional properties but lower caloric content and environmental impact. The replacements are recorded in lists, and a DataFrame summarizing these changes is created.Simulation using the PMA: The PMA simulates the effects of the proposed dietary interventions by optimizing parameters to minimize RMSE in GRU models. These parameters are tailored to each participant’s metabolic profile, enabling accurate predictions of weight changes and metabolic outcomes. Techniques such as Walk-Forward Validation (WFV) and Walk-Forward Simulation (WFS) are employed to validate the model’s predictions.Handling missing data: To maintain data integrity, missing values are addressed using methods from previous studies, ensuring that the dataset remains robust and reliable for simulation purposes.

Following this logic, a comprehensive dictionary was made by compiling a wide range of dietary modifications. This dictionary offers a large collection of personalized food-related adjustments, ranging from portion control to substituting ingredients in ways that reduce both daily calories and environmental impact.

This detailed approach highlights the algorithm’s innovative aspects, focusing on personalized, sustainable dietary interventions that adapt to individual metabolic responses.

All the steps describing this algorithm are resumed in Figure 2.

### 2.4. Validation

Validating the effectiveness of digital biohacking interventions is crucial to understanding their potential impact on weight management and overall well-being. In this section, we delineate our approach to assessing the efficacy of digital biohacking interventions through two distinct analytical phases.

The first phase involves utilizing simulation tools, specifically the Personalized Metabolic Avatar (PMA), to forecast the responses of participants to digital biohacking interventions. By simulating the effects of planned interventions on four participants, we aim to evaluate whether the proposed digital biohacking diet yields tangible improvements in weight loss and overall well-being. This phase transcends mere weight loss outcomes, delving into broader aspects of well-being to comprehensively assess the diet’s efficacy.

In the second phase, we shift our focus to analyzing real-world data to explore the correlation between adherence to digital biohacking interventions and weight loss. By examining actual participant data, we seek to determine whether the adoption of this dietary approach correlates with effective weight reduction and enhanced environmental sustainability.

It is important to note that despite the limited sample size, it is imperative to assess the specific response of each participant. This is because each individual has a unique metabolism, and the algorithm is personalized to each user, rendering each diet plan unique. Understanding the individualized response to digital biohacking interventions allows us to tailor dietary recommendations more effectively, maximizing their impact on weight management and overall health outcomes.

The importance of validating the algorithm using simulation tools lies in its ability to provide insights into potential outcomes before implementation in real-world scenarios. Simulation allows for controlled experimentation, enabling us to forecast and evaluate the effectiveness of digital biohacking interventions without the constraints and uncertainties associated with real-life interventions. By employing simulation tools, we can refine and optimize the algorithm, ensuring its efficacy and relevance in promoting sustainable weight management and overall well-being.

#### 2.4.1. Simulations

In the generation of a biohacked diet, the understanding of individual responses is a critical challenge that must be addressed. Here, we used the PMA to simulate the efficacy of the digital biohacking diet on the four participants. This model was developed, optimizing parameters to minimize Root Mean Square Error (RMSE) in GRU models. These parameters were customized for each participant, enabling the PMA to adapt to their individual metabolic profiles. We extensively assessed the accuracy of weight predictions in a previous study [17], enhancing precision through methodologies such as Walk-Forward Validation (WFV) and Walk-Forward Simulation (WFS). The average RMSE for weight forecasting evaluated was 0.59 ± 0.076, with WFV and WFS averaging 0.42 ± 0.1 and 0.48 ± 0.18, respectively.

We have considered 50 controlled periods of 2 weeks where we applied the digital biohacking approach by using the dictionary made in Section 2.3. To do that, 3 months were considered training for each period. This is made following the walk-forward simulation (WFS) [17] technique, where we used forecasted values of the digital biohacking diet as input rather than actual data. 

To assess whether there is a significant difference between the distribution of weight changes calculated by simulating the digital biohacking diet and the distribution of actual weight changes, a paired *t*-test was performed. In this analysis, we are interested in understanding if the simulated weight changes diverge significantly from the actual ones. This test was chosen because we have paired data, meaning we have corresponding observations of weight changes for the same individuals in both groups. This allows us to directly compare the differences between simulated and actual weight changes for everyone. The *t*-test provides two key insights:-Firstly, it evaluates whether there is a statistically significant difference between the mean weight changes of the two distributions. If the p-value resulting from the *t*-test is below the chosen significance level (α=0.05) we can conclude that there is a significant difference between the distributions. -Secondly, the *t*-test also allows us to investigate the directionality of the difference. By checking the sign of the *t*-statistic, we can determine if the simulated weight changes tend to be lower (statistically negative) or higher than the actual weight changes. A negative *t*-statistic indicates that, on average, the simulated weight changes are lower than the actual weight changes.

By combining the significance test with the directionality analysis, we gain insights into both the statistical difference and the comparative impact of the simulated weight changes. This information helps us to understand whether the digital biohacking diet has a distinct effect on weight loss compared to the actual observations.

#### 2.4.2. Real Data

Then, we investigated the potential of digital biohacking in real-life conditions. To accomplish this, we examined 50 two-week periods for each user, which have different levels of digital biohacking interventions. This one defines how many interventions the digital biohacking suggests in that period. To quantify this index, we calculated the actual total EB (EB_act_) as well as the EB obtained by applying digital biohacking techniques (EB_bh_) to the same period. The difference between these values, referred to as ΔEB = EB_act_ − EB_bh_, quantifies the extent of the difference between the current period’s diet and its biohacked counterpart.

To investigate the relationship between weight changes and ΔEB, we conducted a linear regression analysis, where ΔEB served as the predictor variable (also known as the independent variable, X), while weight changes were considered the outcome variable (or the dependent variable, Y).

Through this analysis, our objective is to confirm if there exists a relationship that confirms the efficacy of periods with fewer interventions on weight loss by determining whether there is a linear association between ΔEB and observed weight changes. This statistical approach allowed us to estimate the slope and intercept of the regression line, enabling us to quantify the relationship, at a personalized level, between these two variables. However, our primary interest lies in verifying whether there is statistical evidence to support the notion that high ΔEB values correspond to an increase in weight gain.

### 2.5. Computational Requirements and Python Libraries

The computational requirements were minimal to allow deployment on virtual machines available on the web. The code for the development of the models was run in Google Colab with the default settings (free plan). The code requires the following libraries: tensorflow = 2.9.2 (https://pypi.org/project/tensorflow/, accessed on 25 January 2024), pandas = 1.3.5 (https://pandas.pydata.org/, accessed on 25 January 2024), numpy = 1.21.6 (https://numpy.org/, accessed on 25 January 2024), matplotlib = 3.2.2 (https://matplotlib.org/, accessed on 25 January 2024), seaborn = 0.11.2 (https://seaborn.pydata.org/, accessed on 25 January 2024), statsmodels = 0.12.2 (https://www.statsmodels.org/stable/index.html, accessed on 25 January 2024), scipy = 1.7.3 (https://pypi.org/project/scipy/, accessed on 25 January 2024), scikit-learn = 1.0.2 (https://scikit-learn.org/stable/, accessed on 25 January 2024) and scikit-posthocs = 0.7.0 (https://scikit-posthocs.readthedocs.io/en/latest/, accessed on 25 January 2024).

## 3. Results

As a starting point, we have generated a personalized digital biohacking diet for each participant. To do that, a substitution list of foods was made by considering more than one year of data. A comprehensive overview of this list was reported in Appendix A for each participant. The application of the full substitutions suggested results in an average reduction of the daily calorie intake of about 14% (average performed on four participants). Moreover, the digital biohacking intervention resulted in a reduction of more than 15% of carbon emissions (Table 1).

### 3.1. Simulations

The PMA enabled us to examine the metabolic responses of each participant to the digital biohacking intervention. To accomplish this, we conducted simulations for 50 two-week periods. In Figure 3, we report for each participant the actual weight changes occurring if no intervention is planned (blue line) and the simulated weight changes occurring during a representative two-week period when participants followed the digital biohacking diet (orange line).

Actual and simulated weight changes averaged over the 50 two weeks period are presented in Table 2, clearly demonstrating the significant impact of digital biohacking in promoting weight loss across all participants when compared to the actual periods.

To further investigate the statistical significance of these outcomes, we conducted a paired *t*-test on the two distributions associated with each participant. The purpose was to determine if there was substantial statistical evidence supporting the differences observed. The results of the test, presented in Table 2, revealed that the *p*-values for each participant were below the predetermined threshold (α = 0.05). This confirms statistically significant differences between the two distributions for each participant. Additionally, the negative *t*-statistic suggests that, on average, the simulated weight changes were lower compared to the actual weight changes.

These results are further graphically displayed in Figure 4, which showcases four bar plots, each representing the distributions of weight variations after two weeks for two groups (a simulation of biohacked periods and without any specific diet).

The bar plots depict the observed weight changes for each user. This finding suggests that the digital biohacking diet may have a noticeable impact on weight changes over the specified time.

### 3.2. Linear Regression Analysis

Given the promising results of the simulations, we have examined the weight difference versus ΔEB calculated over a 50 two-week period for each user to validate the efficacy of digital biohacking (Section 2.4.2) in real-life conditions. In Figure 5, an inverse relationship between the change in ΔEB and weight changes is depicted, as is the retrieved regression line.

There is strong evidence (*p* < 0.05) suggesting that the observed relationship between ΔEB and weight changes is statistically significant (Table 3).

Furthermore, the negative slope of the regression line indicates that, as the ΔEB increases (indicating a larger positive EB), there is a corresponding decrease in weight changes, indicating weight loss.

It is worth noting that the *R*^2^ values reported in Table 3 are relatively low. In linear regression analysis, this value indicates the proportion of variance in the dependent variable (Δw) that can be explained by the independent variable (ΔEB). *R*^2^ values can be influenced by various factors, including the complexity of the relationship, the presence of confounding variables, sample size, and the amount of unexplained variance. Therefore, it is necessary to interpret the *R*^2^ values in conjunction with the statistical significance and direction of the relationship to obtain a comprehensive understanding of the findings. In this case, despite the low *R*^2^ values, the relationship between the digital biohacking diet index and weight changes remains statistically significant due to the reported *p*-value being lower than 0.05.

## 4. Discussion

Obesity and its metabolic complications have emerged as one of the most pressing public health challenges in the 21st century [23]. The prevalence of obesity has reached alarming levels, tripling in many countries across the European Union [24,25,26,27]. The ongoing COVID-19 pandemic has further underscored the significance of addressing obesity, as it has been identified as a risk factor for severe illness [28,29,30]. This has further emphasized the urgent need for effective prevention strategies. Despite extensive research efforts, the debate surrounding the optimal energy content and macronutrient distribution for promoting successful and sustainable weight loss continues [4,5]. Many existing diets offer short-term efficacy, but they often fall short in providing long-term solutions (8). Drastically altering food habits and maintaining strict adherence to specific dietary regimens pose significant challenges [10]. Here, numerous factors come into play, including individual preferences, social influences, and work-related demands, all of which can hinder long-term adherence to a particular diet. To address these challenges and achieve long-lasting results, a comprehensive and lifelong approach is required. This one should incorporate permanent lifestyle changes that seamlessly integrate into individuals’ daily lives. In this context, we have developed an innovative digital biohacking approach to diet, which exploits user data obtained from pre-processed and analyzed information on physical activity, dietary patterns, and anthropometric measurements collected through wearable devices and home-portable technologies. By leveraging this data, we can make minimal yet effective changes to reduce calorie intake and environmental impact. The key aspect of our approach lies in integrating digital biohacking principles into an Internet of Things (IoT)-enabled infrastructure [31,32]. This integration allows for the implementation of simulations and predictions, enabling individuals to gradually adopt healthier habits [16]. By using data-driven insights and personalized recommendations, our innovative diet aims to facilitate sustainable behavior changes and promote healthy weight management. The IoT-reliant infrastructure employed in our approach empowers individuals to take an active role in their own health and well-being [16]. Through real-time feedback, personalized recommendations, and continuous monitoring, individuals can make informed choices and track their progress towards achieving their weight loss and overall health goals.

In this manuscript, we have presented compelling evidence supporting the effectiveness of digital biohacking as a viable solution for reducing calorie intake and facilitating weight loss through minimal yet precise modifications to the diet.

By simulating 50 two-week weight variation periods for the four participants using the PMA, we have compared the simulated Δw with their actual ones. This approach allowed us to assess the effectiveness of the digital biohacking interventions in driving weight loss and provided valuable insights into the potential benefits of adopting digital biohacking strategies.

To further evaluate the efficacy of the biohacked diets, a linear regression analysis was carried out by analyzing the individual data of the four participants. This analysis provided insights into the direction and magnitude of the relationship between real periods and different levels of digital biohacking interventions for everyone. The negative slope observed suggests that, on average, a period that requires fewer interventions with respect to others was associated with weight loss among the participants. These findings indicate that periods closest to their digital biohacking version (i.e., the digital biohacking applied to those periods) have a significant impact on weight loss.

One of the key strengths of digital biohacking lies in its ability to consider various user data, including personal taste preferences, daily caloric intake, meal structure, and more, to make minimal changes that seamlessly fit into their daily lives. Unlike generic diet plans, our approach generated four distinct, personalized diets tailored to everyone. As highlighted in Table 4, generic diets inherently lack the capacity for true personalization, while digital biohacking enables the creation of personalized food choices for users.

Another notable advantage of digital biohacking is its contribution to reducing emissions. As reported in Appendix A, food consumption accounts for a substantial portion of total emissions, approximately 26% (Reference [33] is cited in the Appendix A). Consequently, addressing this pollution challenge [34] becomes of utmost importance. Through digital biohacking, an average reduction of around 15% (Table 2) can be achieved, making a substantial contribution to the overarching goal of minimizing emissions and aligning with the current emphasis on environmentally friendly practices. In the face of the urgent and immediate challenge of climate change, it is imperative to address this issue across all sectors. By embracing digital biohacking techniques and making sustainable dietary choices, individuals have the power to actively reduce their carbon footprint and play a part in mitigating the detrimental effects of greenhouse gas emissions on our planet.

Therefore, digital biohacking holds the potential to serve as a powerful support tool for nutritionists, dietitians, and other professionals. It can lay the foundation for truly personalized nutrition approaches with long-term results, leveraging real data to identify minimal changes that facilitate weight loss and emission reduction while prioritizing overall well-being. Generated diet and activity plans can be delivered to users through front-end components, supported by virtual assistants to monitor behavior and enhance adherence to optimal actions. The innovation of this approach lies in the use of the PMA to tailor dietary interventions to individual metabolic profiles. The algorithm’s systematic process of food replacement, combined with advanced simulation techniques and robust handling of missing data, distinguishes it from traditional dietary models. By integrating personalized data inputs, the PMA ensures that dietary recommendations are not only effective but also sustainable and minimally disruptive to the user’s existing dietary habits. These innovations underscore the potential of digital biohacking as a powerful tool for personalized nutrition and sustainable weight management.

However, it is important to acknowledge the critical issues associated with digital biohacking. Firstly, as the changes are data-driven, there may be limitations in critical situations, such as extremely high caloric alimentation. Additionally, the analysis of digital biohacking effects revealed varying outcomes among the four participants, based on their individual conditions. While two participants experienced significant weight loss, another maintained their weight, and the last participant showed a reduction in weight gain (Figure 4). Despite this, in all cases, the application of digital biohacking interventions yielded a significant weight reduction compared to the absence of any intervention.

To address these challenges, digital biohacking could benefit from incorporating pre-modification of the diet, allowing for gradual improvements in alimentation. Furthermore, at the current stage of development, digital biohacking requires at least two months of data collection to achieve optimal performance. To enhance data collection, the evolving field of automatic food detection methods through mobile phones [35] could alleviate the burden on users by reducing manual data compilation.

We recognize that the study’s small sample size, limited gender diversity, and inclusion of predominantly normal BMI and sedentary individuals may impact the generalizability of our findings. However, this pilot study was designed as a proof-of-concept to establish the feasibility and accuracy of our digital biohacking approach. Our initial focus on a controlled cohort allows us to make precise adjustments and improvements to the algorithm before testing it in larger, more varied populations. Future research will aim to include participants with a wider range of BMIs, more balanced gender representation, and varying levels of physical activity to enhance the external validity of our findings. It is important to emphasize that this study, although conducted on a small cohort of just four participants, does not focus primarily on clinical outcomes. Instead, it highlights the procedural advancements and methodological insights gained, particularly in using a PMA to simulate the impact of diet on weight management.

## 5. Limitations and Future Trends

Despite the promising potential of digital biohacking for weight management, several limitations and future directions must be considered for its optimization and validation. While this study introduces a novel and optimized dietary approach based on the well-established principle of caloric deficit [36], the sample size is insufficient to statistically validate this new methodology, which is applicable to other types of diets as well. Therefore, future research should involve larger populations to enhance the generalizability of the findings.

Another major challenge is the burden of data collection. Accurate and continuous data input is crucial for personalizing the dietary recommendations effectively. This requires users to consistently track their dietary intake, physical activity, and other relevant health metrics. Such detailed tracking can be time-consuming and may lead to user fatigue over time, potentially reducing adherence.

Additionally, maintaining adherence to the biohacking diet over longer periods of time can be challenging. The success of this approach relies on users making incremental changes and consistently following through with them. Social and environmental factors, such as changes in routine, social events, or limited access to recommended foods, can disrupt adherence. Ensuring long-term commitment requires ongoing motivation and support, which may involve regular feedback, adjustments to the diet plan, and possibly even social or community-based support systems.

Furthermore, there is a need for robust data privacy and security measures, given the sensitive nature of health-related information being collected and analyzed. Users must trust that their data will be handled securely and that their privacy will be protected to be willing to participate fully.

By addressing these challenges, we aim to improve the feasibility and effectiveness of the digital biohacking approach, ensuring it provides a sustainable and personalized solution for long-term weight management and health improvement.

## 6. Conclusions

This study underscores the potential of digital biohacking as a highly effective and personalized approach for reducing calorie intake, promoting weight loss, and addressing environmental impact. By leveraging real data and implementing precise yet minimal modifications, digital biohacking emerges as a promising paradigm in the field of nutrition and sustainable dietary practices. Its integration could empower nutritionists and dietitians with scientific tools and validated instruments, enabling the development of diets that yield long-lasting results in the pursuit of a healthy lifestyle. To further enhance the effectiveness of personalized diets, the integration of advanced techniques such as deep learning, specifically Generative Adversarial Networks (GANs) [37], could be considered. This integration would optimize the generation of personalized diets, thereby increasing their effectiveness across a wider range of cases. Additionally, incorporating microbiome analysis [38,39] could play a crucial role in developing diets that have a positive impact on gut microbiota health. Furthermore, digital biohacking offers significant potential beyond calorie reduction alone. It facilitates comprehensive and sustainable dietary modifications, including the optimization of macronutrient distribution, generating personalized diets that can be tailored to address specific nutritional needs, and ensuring a well-balanced intake of proteins, carbohydrates, and fats. This approach not only promotes weight loss but also fosters long-term health and well-being. Another crucial point that digital biohacking could address is the reduction of salt intake, which is a critical aspect of a healthy diet [40]. Excessive sodium consumption is associated with various health risks, such as hypertension and cardiovascular diseases [41]. By employing data-driven strategies, digital biohacking can allow for the customization of diets that can gradually reduce salt levels while preserving flavor and satisfaction. This personalized approach encourages individuals to adopt a sustainable and healthier eating pattern. Moreover, the integration of innovative anthropometric markers, tracked through wearable devices, such as VO^2^ max [42,43] and heart rate variability (HRV) [44], shows potential for generating diets aimed at enhancing these performance metrics. Additionally, the incorporation of newly developed biomarkers of lipid metabolism, such as membrane lipids and membrane fluidity of red blood cells, enables the formulation of diets that consider the effects and impacts of dietary molecules on these outcomes [45,46]. By leveraging these advancements, digital biohacking can not only contribute to personal health but also promote a reduction in emissions associated with food consumption.

In conclusion, while this pilot study provides critical insights and initial validation of our digital biohacking methodology, we acknowledge its limitations regarding sample size and demographic diversity. Our findings serve as a foundational step towards developing a robust, personalized dietary intervention tool. Subsequent research will involve larger, more diverse cohorts to fully assess the scalability and effectiveness of our approach across different population segments.

## Figures and Tables

**Figure 1 nutrients-16-02021-f001:**
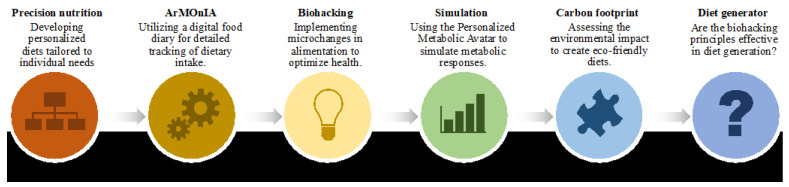
Study framework diagram. This diagram illustrates the comprehensive workflow of the study, highlighting key components at each stage: Precision Nutrition: Developing personalized diets tailored to individual needs. ArMOnIA: Utilizing a digital food diary for detailed tracking of dietary intake. Biohacking: Implementing microchanges in alimentation to optimize health. Simulation (PMA): Using the Personalized Metabolic Avatar to simulate metabolic responses. Carbon Footprint: Assessing the environmental impact to create eco-friendly diets. Diet Generator: Evaluating the effectiveness of biohacking principles in diet generation.

**Figure 2 nutrients-16-02021-f002:**
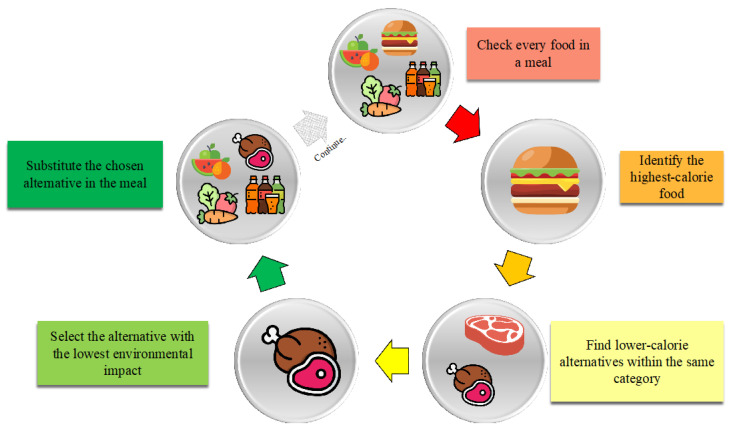
The figure displays the block diagram of the algorithm underlying digital biohacking.

**Figure 3 nutrients-16-02021-f003:**
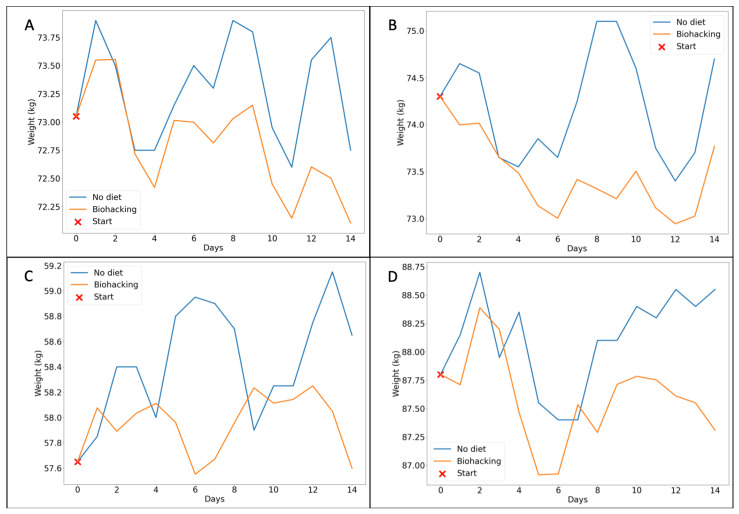
The figure presents the simulation of a representative two-week period by applying digital biohacking interventions for the four participants (**A**–**D**) using the PMA. In this context, day 0 denotes the beginning of the diet implementation.

**Figure 4 nutrients-16-02021-f004:**
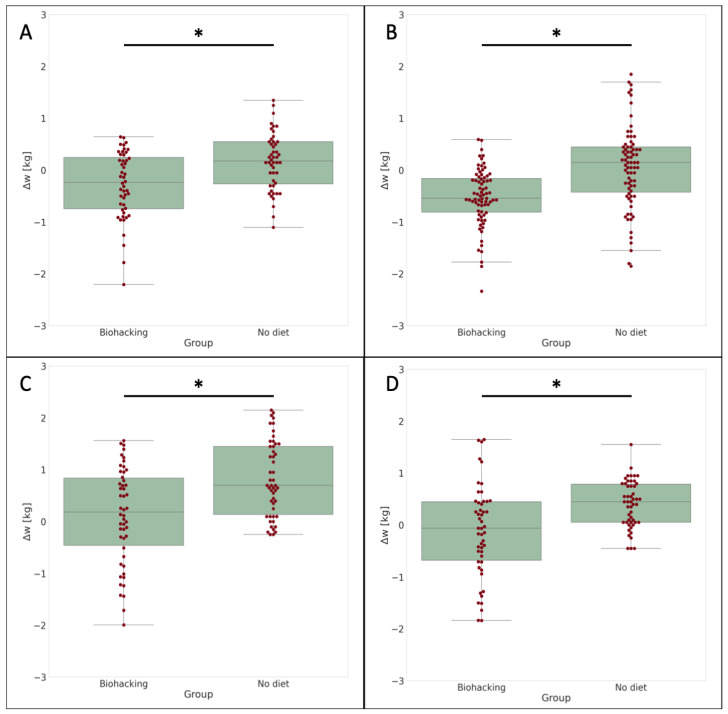
The figure illustrates the distributions of Δw in two groups: digital biohacking and No Diet. The distributions are presented individually for each participant (**A**–**D**). In the figure, the symbol “*” denotes a *p*-value that is lower than 0.05, indicating a statistically significant difference between the groups.

**Figure 5 nutrients-16-02021-f005:**
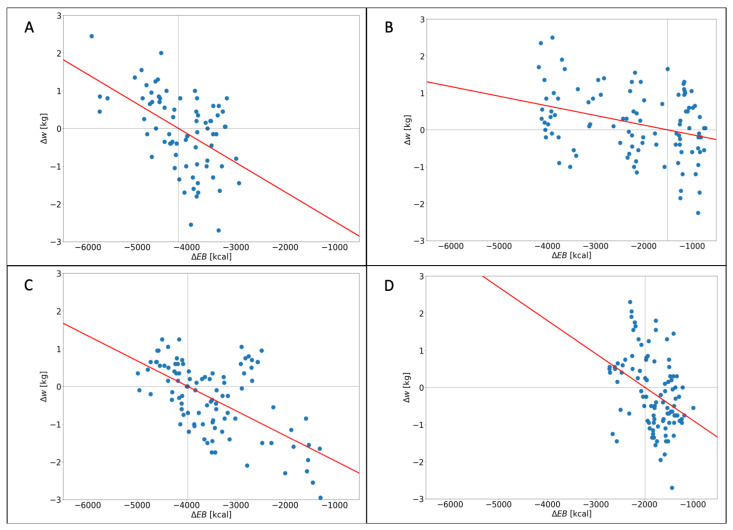
The figure shows the relationship between the distribution of ΔEB and the corresponding Δw for each participant. Additionally, a red linear trend, derived from the results of a linear regression analysis, is displayed to illustrate the overall relationship between the variables. Each subplot represents data for a specific user: (**A**) for user 0, (**B**) for user 1, (**C**) for user 2, and (**D**) for user 3.

**Table 1 nutrients-16-02021-t001:** Average effects of digital biohacking interventions ^1^.

Parameter	Average Daily Variation	Decrease Percentage
Average daily intake reduction	−236.78 ± 50.65 kcal	14.24 ± 3.1%
Average daily carbon footprint impact reduction	$-736.48\;\pm \;146\;gCO_{2}eq$	15.12 ± 1.13%

^1^ The table provides a comprehensive summary of the average impact of the digital biohacking diet on the four participants, expressed as reductions in caloric intake and carbon footprint impact. These values are presented along with their respective standard deviations and are also expressed as percentages.

**Table 2 nutrients-16-02021-t002:** Comparison of Simulated digital biohacking Effects vs. Real Data: Weight Variation ^1^.

Participant	Actual Delta Weight	Simulated Delta Weight	Delta Weight Loss ^2^	*p*-Value ^3^	*t*-Statistic ^4^
0	0.17 ± 0.54 kg	−0.29 ± 0.64 kg	−0.48 ± 0.54 kg	$5.18\times 10^{-7}$	−5.81
1	−0.02 ± 0.86 kg	−0.70 ± 0.56 kg	−0.68 ± 0.77 kg	$2.43\times 10^{-8}$	−6.24
2	0.82 ± 0.72 kg	0.12 ± 0.92 kg	−0.70 ±1.06 kg	$2.58\times 10^{-5}$	−4.64
3	0.40 ± 0.44 kg	−0.12 ± 0.87 kg	−0.52 ± 1.01 kg	$0.71\times 10^{-3}$	−3.62

^1^ The table provides a comprehensive compilation of weight changes obtained from simulations conducted using the PMA during periods of digital biohacking implementation, as well as the actual observed weight changes and the results of the paired *t*-test. ^2^ The corresponding weight loss values resulting from these two conditions, specifically the difference between simulated delta weight and actual delta weight, are calculated and presented in the column labeled “Delta Weight Loss”. ^3^ Each participant’s *p*-value is below the specified threshold (α = 0.05), indicating that digital biohacking has a significant effect on weight changes compared to the no-diet condition. ^4^ The negative values of the *t*-statistic further ensure that digital biohacking leads to weight loss compared to the no-diet conditions.

**Table 3 nutrients-16-02021-t003:** Linear regression results ^1^.

Participant	*p*-Value ^2^	Slope (Kg/Kcal) ^3^	$R^{2}$ ^4^	Pearson Coefficient ^5^
0	$2.99\times 10^{-6}$	−0.0008	0.25	−0.5
1	$8.86\times 10^{-4}$	−0.0003	0.11	−0.33
2	$1.2\times 10^{-4}$	−0.0009	0.14	−0.37
3	$6.89\times 10^{-12}$	−0.0007	0.38	−0.62

^1^ The table presents the results of a linear regression analysis conducted on a distribution of ΔEB, representing the difference between real periods and their biohacked versions, alongside the corresponding weight changes. ^2^ The *p*-value reflects the statistical significance of the relationship between the ΔEB and weight changes. In this analysis, they are below the threshold of 0.05, indicating a statistically significant relationship. ^3^ The slope in this context represents the rate of change in weight per unit change in the ΔEB. The negative slopes observed indicate that higher ΔEB values are associated with weight gain. ^4^ The $R^{2}$ values provide insights into the proportion of variation in weight changes that can be explained by the ΔEB. The relatively lower $R^{2}$ values suggest that the ΔEB alone may not account for a substantial proportion of the variation in weight changes. ^5^ The Pearson coefficient, ranging from −1 to +1, measures the strength and direction of the linear relationship between the ΔEB and weight changes.

**Table 4 nutrients-16-02021-t004:** Comparison between generic diets and digital biohacking ^1^.

	Generic Diets	Digital Biohacking
Personalization	Limited	Highly Customized
User-specific data consideration	Minimal	Comprehensive
Incorporation of taste preferences	Limited	Extensive
Sustainability (emission reduction)	Not addressed	Addressed
Long-term adherence potential	Challenging	Promising

^1^ The table provides an overview of the key strengths of digital biohacking in comparison to generic diets. Specifically, the digital biohacking diet offers the ability to create a personalized, user-specific, sustainable, and effective long-term dietary approach.

## Data Availability

The code for the biohacking algorithm is available at the following link: https://github.com/bio-hacking-tech/biohacking/tree/main, accessed on 25 June 2024.

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
