# Peer review of "Digital Biohacking Approach to Dietary Interventions: A Comprehensive Strategy for Healthy and Sustainable Weight Loss"

_nutrients, 2024, doi:10.3390/nu16132021_

Round 1
Reviewer 1 Report (Previous Reviewer 2)
Comments and Suggestions for Authors
1- Provide more details on the participant demographics (age range, BMI, etc.) and recruitment process in the Methods section.
2- Expand the limitations of the small sample size and the need for larger, controlled studies to validate the findings. I know that you have already mentioned it, but further explaining and expanding it would be optimal.
3- Expand on the potential challenges of implementing digital biohacking diets, such as data collection burden and adherence over longer periods.
4- Clarify how the personalized metabolic model (PMA) was developed and validated.
Last of all, if you could add a figure illustrating the digital biohacking algorithm workflow for better reader understanding, that would be optimal, however, I don't think this would be absolutely necessary for publication.
Author Response
1- Provide more details on the participant demographics (age range, BMI, etc.) and recruitment process in the Methods section.
Thank you for your suggestion. I have included additional details on the participant demographics in the Methods section, as follows (Page 2, Line 72):
"””
In this single-arm, uncontrolled-pilot prospective study, a group of 4 voluntary healthy Italian adults with sedentary lifestyles (25% females and 75% males, 1 overweight and 3 normal, BMI of 23.42±1.68, age of 44.75±10.23) was recruited.
“”"
2- Expand the limitations of the small sample size and the need for larger, controlled studies to validate the findings. I know that you have already mentioned it, but further explaining and expanding it would be optimal.
We understand the importance of discussing broader implications and limitations. Therefore, we have included the following discussion points in the manuscript (Page 13, Line 465):
“”"
We recognize that the small sample size, limited gender diversity (predominantly male), and inclusion of only sedentary individuals may affect the generalizability of our study's findings. Nevertheless, this sample offers valuable preliminary data, especially useful for understanding metabolic responses in a sedentary adult population. The BMI and age range suggest a middle-aged group with relatively healthy weight statuses, which can help establish baseline expectations for similar demographic profiles in future, larger studies.
It's important to emphasize that this study, although conducted on a small cohort of just four participants, does not focus primarily on clinical outcomes. Instead, it highlights the procedural advancements and methodological insights gained, particularly in using a Personalized Metabolic Avatar to simulate the impact of diet on weight management.
“”"
Additionally, we have incorporated a paragraph addressing the limitations and future trends (Page 13, Line 476):
“””
- Limitations and future trends
Despite the promising potential of digital biohacking for weight management, several limitations and future directions must be considered for its optimization and validation. While this study introduces a novel and optimized dietary approach based on the well-established principle of caloric deficit, the sample size is insufficient to statistically validate this new methodology. Therefore, future research should involve larger populations to enhance the generalizability of the findings.
Another major challenge is the burden of data collection. Accurate and continuous data input is crucial for personalizing the dietary recommendations effectively. This requires users to consistently track their dietary intake, physical activity, and other relevant health metrics. Such detailed tracking can be time-consuming and may lead to user fatigue over time, potentially reducing adherence.
Additionally, maintaining adherence to the biohacking diet over longer periods can be challenging. The success of this approach relies on users making incremental changes and consistently following through with them. Social and environmental factors, such as changes in routine, social events, or limited access to recommended foods, can disrupt adherence. Ensuring long-term commitment requires ongoing motivation and support, which may involve regular feedback, adjustments to the diet plan, and possibly even social or community-based support systems.
Furthermore, there is a need for robust data privacy and security measures, given the sensitive nature of health-related information being collected and analyzed. Users must trust that their data will be handled securely and that their privacy will be protected to be willing to participate fully.
By addressing these challenges, we aim to improve the feasibility and effectiveness of the digital biohacking approach, ensuring it provides a sustainable and personalized solution for long-term weight management and health improvement.
“”"
3- Expand on the potential challenges of implementing digital biohacking diets, such as data collection burden and adherence over longer periods.
Thank you for your feedback. I have expanded on the limitations and potential challenges of implementing digital biohacking diets in the manuscript. The added paragraph is now at Page 13, Line 476.
4- Clarify how the personalized metabolic model (PMA) was developed and validated.
Thank you for your inquiry. We have clarified how the personalized metabolic model (PMA) was developed and validated in the manuscript. The added text can be now found at Page 5, Line 199:
“””
This model was developed, optimizing parameters to minimize Root Mean Square Error (RMSE) in GRU models. These parameters were customized for each participant, enabling the PMA to adapt to their individual metabolic profiles. We extensively assessed the accuracy of weight predictions in a previous study [17], enhancing precision through methodologies such as Walk-Forward Variation (WFV) and Walk-Forward Simulation (WFS). The average RMSE for weight forecasting evaluated was 0.59 ± 0.076, with WFV and WFS averaging 0.42 ± 0.1 and 0.48 ± 0.18, respectively.
“””
Last of all, if you could add a figure illustrating the digital biohacking algorithm workflow for better reader understanding, that would be optimal, however, I don't think this would be absolutely necessary for publication.
Thank you for your thoughtful suggestion regarding the inclusion of an additional figure to illustrate the digital biohacking algorithm workflow. We appreciate your consideration of enhancing reader understanding. After careful consideration, we think that the existing Figure 1 adequately depicts the workflow of the model, providing a clear visualization for our readers. We are confident that Figure 1 effectively communicates the essential aspects of the digital biohacking algorithm and its implementation. Therefore, we respectfully propose to maintain Figure 1 as the sole illustration of the workflow in our manuscript. We hope this decision aligns with your expectations, and we remain open to further discussion or clarification if needed.

Reviewer 2 Report (New Reviewer)
Comments and Suggestions for Authors
This research seeks to create and assess a new digital biohacking method for making dietary changes that promote sustainable weight loss and decrease carbon footprint impact. A small-scale trial involving four individuals was carried out over a year. Information was gathered on their food consumption, carbon footprint, calorie intake, macronutrient balance, weight, and energy expenditure. The approach utilized a digital model of metabolism called the Personalized Metabolic Avatar (PMA) to mimic weight fluctuations and to strategize and implement digital interventions for dietary adjustments.
It is a quite interesting and significant paper. The Authors have presented sufficient data. The appropriate tables and figures have been provided. The article is easy to read and logically structured. The methods are adequately described. The Authors used proper statistical methods. The conclusions are consistent with the presented evidence and arguments.
I have some remarks, which should be taken into consideration:
1. What are the limitations of your study?
2. Please add the sections: Limitations and Future trends
3. Please add strengths to the discussion section
Author Response
This research seeks to create and assess a new digital biohacking method for making dietary changes that promote sustainable weight loss and decrease carbon footprint impact. A small-scale trial involving four individuals was carried out over a year. Information was gathered on their food consumption, carbon footprint, calorie intake, macronutrient balance, weight, and energy expenditure. The approach utilized a digital model of metabolism called the Personalized Metabolic Avatar (PMA) to mimic weight fluctuations and to strategize and implement digital interventions for dietary adjustments.
It is a quite interesting and significant paper. The Authors have presented sufficient data. The appropriate tables and figures have been provided. The article is easy to read and logically structured. The methods are adequately described. The Authors used proper statistical methods. The conclusions are consistent with the presented evidence and arguments.
I have some remarks, which should be taken into consideration:
- What are the limitations of your study?
- Please add the sections: Limitations and Future trends
Both suggestions have been implemented. We have added a paragraph under the heading "Limitations and future trends" (Page 13, Line 476) to address the limitations of the study and provide insights into future research directions. Thank you for your feedback.
“””
- Limitations and future trends
Despite the promising potential of digital biohacking for weight management, several limitations and future directions must be considered for its optimization and validation. While this study introduces a novel and optimized dietary approach based on the well-established principle of caloric deficit, the sample size is insufficient to statistically validate this new methodology. Therefore, future research should involve larger populations to enhance the generalizability of the findings.
Another major challenge is the burden of data collection. Accurate and continuous data input is crucial for personalizing the dietary recommendations effectively. This requires users to consistently track their dietary intake, physical activity, and other relevant health metrics. Such detailed tracking can be time-consuming and may lead to user fatigue over time, potentially reducing adherence.
Additionally, maintaining adherence to the biohacking diet over longer periods can be challenging. The success of this approach relies on users making incremental changes and consistently following through with them. Social and environmental factors, such as changes in routine, social events, or limited access to recommended foods, can disrupt adherence. Ensuring long-term commitment requires ongoing motivation and support, which may involve regular feedback, adjustments to the diet plan, and possibly even social or community-based support systems.
Furthermore, there is a need for robust data privacy and security measures, given the sensitive nature of health-related information being collected and analyzed. Users must trust that their data will be handled securely and that their privacy will be protected to be willing to participate fully.
By addressing these challenges, we aim to improve the feasibility and effectiveness of the digital biohacking approach, ensuring it provides a sustainable and personalized solution for long-term weight management and health improvement.
“”"
- Please add strengths to the discussion section
Thank you for your suggestion. We have revised the discussion section to highlight the strengths of digital biohacking, emphasizing its personalized approach, tailored interventions, focus on sustainability and emission reduction, as well as its potential as a valuable support tool for professionals in the field (page 12 lines 422).
Reviewer 3 Report (New Reviewer)
Comments and Suggestions for Authors
This study is very interesting. However, the data was from a rather small sample of 4 people, and further research is needed to draw reliable conclusions. Therefore, at this point, I think it is reasonable to reject the application.
Author Response
This study is very interesting. However, the data was from a rather small sample of 4 people, and further research is needed to draw reliable conclusions. Therefore, at this point, I think it is reasonable to reject the application.
We understand the importance of discussing broader implications and limitations. Therefore, we have included the following discussion points in the manuscript (Page 13, Line 465):
“”"
We recognize that the small sample size, limited gender diversity (predominantly male), and inclusion of only sedentary individuals may affect the generalizability of our study's findings. Nevertheless, this sample offers valuable preliminary data, especially useful for understanding metabolic responses in a sedentary adult population. The BMI and age range suggest a middle-aged group with relatively healthy weight statuses, which can help establish baseline expectations for similar demographic profiles in future, larger studies.
It's important to emphasize that this study, although conducted on a small cohort of just four participants, does not focus primarily on clinical outcomes. Instead, it highlights the procedural advancements and methodological insights gained, particularly in using a Personalized Metabolic Avatar to simulate the impact of diet on weight management.
“”"
Additionally, we have incorporated a paragraph addressing the limitations and future trends (Page 13, Line 476):
“””
- Limitations and future trends
Despite the promising potential of digital biohacking for weight management, several limitations and future directions must be considered for its optimization and validation. While this study introduces a novel and optimized dietary approach based on the well-established principle of caloric deficit, the sample size is insufficient to statistically validate this new methodology. Therefore, future research should involve larger populations to enhance the generalizability of the findings.
Another major challenge is the burden of data collection. Accurate and continuous data input is crucial for personalizing the dietary recommendations effectively. This requires users to consistently track their dietary intake, physical activity, and other relevant health metrics. Such detailed tracking can be time-consuming and may lead to user fatigue over time, potentially reducing adherence.
Additionally, maintaining adherence to the biohacking diet over longer periods can be challenging. The success of this approach relies on users making incremental changes and consistently following through with them. Social and environmental factors, such as changes in routine, social events, or limited access to recommended foods, can disrupt adherence. Ensuring long-term commitment requires ongoing motivation and support, which may involve regular feedback, adjustments to the diet plan, and possibly even social or community-based support systems.
Furthermore, there is a need for robust data privacy and security measures, given the sensitive nature of health-related information being collected and analyzed. Users must trust that their data will be handled securely and that their privacy will be protected to be willing to participate fully.
By addressing these challenges, we aim to improve the feasibility and effectiveness of the digital biohacking approach, ensuring it provides a sustainable and personalized solution for long-term weight management and health improvement.
“”"
Round 2
Reviewer 3 Report (New Reviewer)
Comments and Suggestions for Authors
I have confirmed that the issues I pointed out last time are comprehensively described mainly in the discussion section. As I pointed out last time, I understand that this is a very interesting study, but I think that the number of samples should be increased before publishing.
Author Response
I have confirmed that the issues I pointed out last time are comprehensively described mainly in the discussion section. As I pointed out last time, I understand that this is a very interesting study, but I think that the number of samples should be increased before publishing.
We acknowledge the limitation posed by our sample size; however, our study is built upon well-established metabolic dynamics, where a caloric deficit is known to induce weight loss. Indeed, our study aims to explore the transformative potential of digital applications in the realm of nutrition, which have facilitated the retrieval of highly personalized dietary histories. This data is integral within the framework of precision nutrition. Our study serves as a pilot investigation, wherein we delve into the application of biohacking principles leveraging this data, and subsequently test them using metabolic models capable of simulating metabolic responses. In this context, both the dietary interventions and the metabolic models are intricately personalized, tailored to individual users.
To elucidate the rationale behind our study, we have provided the following detailed explanation:
Revised Manuscript Sections:
Introduction (to include rationale for study design):
New Text: (Line 70-77 Pag. 2)
"In this pilot study, we aimed to test the feasibility and initial effectiveness of a novel digital biohacking approach to dietary interventions. While acknowledging the limited sample size and demographic constraints, we deliberately selected a small, controlled cohort to rigorously test and refine our methodology. This initial cohort comprised predominantly normal BMI individuals to ensure a controlled baseline for evaluating the algorithm's precision and adaptability in a sedentary population. Furthermore, focusing on a limited number of participants allowed us to closely monitor and accurately measure a wide array of variables necessary for developing our Personalized Metabolic Avatar (PMA)."
Methods (to emphasize rationale and future directions):
Expanded Text: (Line 80-95, Pag. 3)
"In this single-arm, uncontrolled pilot prospective study, our sample comprised four voluntary healthy Italian adults with sedentary lifestyles (25% females and 75% males). The mean BMI of participants was 23.42±1.68, and their average age was 44.75±10.23 years. One participant was classified as overweight while the remaining three had normal weight status. They were recruited among our lab staff, self-monitored their weight, diet and activities completed for more than 1 year, daily, as already shown in previous works, under the surveillance of a nutritionist [17,18]. The participants shared their personal data after signing informed consent. This specific demographic was chosen to provide a controlled baseline, ensuring that the effects of the digital biohacking interventions could be measured with minimal confounding variables. Although this cohort is not representative of the broader population, it provides valuable preliminary data for refining our methodology. Future studies will aim to include larger, more diverse populations to validate and generalize our findings."
Discussion (to address limitations and future research):
New Text: (Line 492-499, Pag. 14)
"We recognize that the study’s small sample size, limited gender diversity, and inclusion of predominantly normal BMI, sedentary individuals may impact the generalizability of our findings. However, this pilot study was designed as a proof of concept to establish the feasibility and accuracy of our digital biohacking approach. Our initial focus on a controlled cohort allows us to make precise adjustments and improvements to the algorithm before testing it in larger, more varied populations. Future research will aim to include participants with a wider range of BMIs, more balanced gender representation, and varying levels of physical activity to enhance the external validity of our findings."
Conclusion (to reiterate the study's scope and future plans):
New Text: (Line 563-569, Pag. 15)
"In conclusion, while this pilot study provides critical insights and initial validation of our digital biohacking methodology, we acknowledge its limitations regarding sample size and demographic diversity. Our findings serve as a foundational step towards developing a robust, personalized dietary intervention tool. Subsequent research will involve larger, more diverse cohorts to fully assess the scalability and effectiveness of our approach across different population segments."
This manuscript is a resubmission of an earlier submission. The following is a list of the peer review reports and author responses from that submission.
Round 1
Reviewer 1 Report
Comments and Suggestions for Authors The present study deals with the development and evaluation of a digital tool suggesting dietary modifications. The main question addressed by the research is weather a digital tool which is designed to select foods with the highest calorie intake in each meal and replace them with healthier foods taken from personal food diaries can be able to promote weight loss and reduce carbon footprint impact in 4 subjects. The authors concluded that dietary modifications suggested by the digital tool was able to reduce by ~14% daily calorie reduction and by ~15% carbon footprint impact. Comments: Concept: The idea is original and the topic is interesting and important since obesity is increasing worldwide. Introduction: The introduction presents the background in the topic. Study design: A major concern about the study is the sample size. In materials and methods section there is no information about how the authors have calculated the appropriated number of participants in the study. This study has been conducted in a very small sample size (4 subjects). Moreover, the sample size is heterogenic in terms of sex and body weight (1 females and 3 males, 1 overweight and 3 normal). Moreover, it could be useful to have information about subjects characteristics in terms of demographic data, co-morbidities and medication use. In this reviewers' point of view the sample size has to be increased in order to increase the validity of the data. Besides validity, an increased sample size with an adequate group of overweight/obese and lean subjects could allow statistical analysis to be conducted between subgroups. Conclusions: The authors conclude that the certain digital tool seems promising in combating obesity while considering both individual preferences and environmental sustainability. It is questionable whether experiments conducted in one obese subject can be considered adequate to draw conclusions about the management of obesity. The same stands about sex. A study conducted in only one woman is questionable whether it can lead to conclusions that can be generalized in both sexes. Discussion: The first part of the discussion presents a general view on the background of the topic which is also presented in the introduction section. The quality of figures is also acceptable. References are appropriate.
Reviewer 2 Report
Comments and Suggestions for Authors
1- Expand on the limitations of the study, especially the small sample size and potential biases in self-reporting.
2- Provide clearer explanations of the digital biohacking process and Personalized Metabolic Avatar.
3- Include more detailed information on participant recruitment and selection criteria.
4- Address the generalizability of the results to broader populations.